# Fetal Hyperthyroidism with Maternal Hypothyroidism: Two Cases of Intrauterine Therapy

**DOI:** 10.3390/diagnostics14010102

**Published:** 2024-01-03

**Authors:** Lu Hong, Mary Hoi Yin Tang, Ka Wang Cheung, Libing Luo, Cindy Ka Yee Cheung, Xiaoying Dai, Yanyan Li, Chuqin Xiong, Wei Liang, Wei Xiang, Liangbing Wang, Kelvin Yuen Kwong Chan, Shengmou Lin

**Affiliations:** 1Prenatal Diagnosis Centre, The University of Hong Kong—Shenzhen Hospital, Shenzhen 518053, China; 2Department of Obstetrics and Gynaecology, Queen Mary Hospital, The University of Hong Kong, Hong Kong 999077, China; 3Department of Ultrasound, The University of Hong Kong—Shenzhen Hospital, Shenzhen 518053, China; 4Department of Endocrinology, The University of Hong Kong—Shenzhen Hospital, Shenzhen 518053, China; 5Neonatal Intensive Care Unit, The University of Hong Kong—Shenzhen Hospital, Shenzhen 518053, China; 6Department of Applied Science, School of Science and Technology, Hong Kong Metropolitan University, Hong Kong 999077, China

**Keywords:** fetal hyperthyroidism, thyroid-stimulating hormone receptor antibody (TRAb), intrauterine therapy

## Abstract

Fetal hyperthyroidism can occur secondary to maternal autoimmune hyperthyroidism. The thyroid-stimulating hormone receptor antibody (TRAb) transferred from the mother to the fetus stimulates the fetal thyroid and causes fetal thyrotoxicosis. Fetuses with this condition are difficult to detect, especially after maternal Graves disease therapy. Here, we present two cases of fetal hyperthyroidism with maternal hypothyroidism and review the assessment and intrauterine therapy for fetal hyperthyroidism. Both women were referred at 22^+^ and 23^+^ weeks of gestation with abnormal ultrasound findings, including fetal heart enlargement, pericardial effusion, and fetal tachycardia. Both women had a history of Graves disease while in a state of hypothyroidism with a high titer of TRAb. A sonographic examination showed a diffusely enlarged fetal thyroid with abundant blood flow. Invasive prenatal testing revealed no significant chromosomal aberration. Low fetal serum TSH and high TRAb levels were detected in the cord blood. Fetal hyperthyroidism was considered, and maternal oral methimazole (MMI) was administered as intrauterine therapy, with the slowing of fetal tachycardia, a reduction in fetal heart enlargement, and thyroid hyperemia. During therapy, maternal thyroid function was monitored, and the dosage of maternal levothyroxine was adjusted accordingly. Both women delivered spontaneously at 36^+^ weeks of gestation, and neonatal hyperthyroidism was confirmed in both newborns. After methimazole and propranolol drug treatment with levothyroxine for 8 and 12 months, both babies became euthyroid with normal growth and development.

## 1. Introduction

Fetal hyperthyroidism is a rare and life-threatening disease. This disorder can be classified as autoimmune and non-autoimmune hyperthyroidism. The former usually occurs secondary to maternal autoimmune hyperthyroidism mediated by autoantibodies. The thyroid-stimulating hormone receptor antibody (TRAb) is transferred from the mother to the fetus through the placenta, stimulating the fetal thyroid and resulting in fetal thyrotoxicosis. The clinical manifestations of fetal thyrotoxicosis include fetal goiter, tachycardia, intrauterine growth restriction, non-immune hydrops, accelerated bone maturation, and intrauterine death [1].

Overt hyperthyroidism occurs in 0.1% to 0.4% of all pregnancies [2]. Subclinical hyperthyroidism in the first trimester persists until the third trimester in 21% of cases, as reported by a Chinese study of 42,492 mothers [3], and 1–5% of children born to these women have hyperthyroidism [4,5,6]. For women with current or past Graves disease, assays to measure TRAb should be performed at the beginning of pregnancy with fetal ultrasound surveillance for the early detection of fetal hyperthyroidism [7]. Several reports [8,9,10] describe the treatment of fetal congenital hyperthyroidism by giving mothers antithyroid drugs. Here, we present two cases of fetal hyperthyroidism with maternal hypothyroidism and review the assessment and intrauterine therapy of fetal hyperthyroidism.

## 2. Case Presentation

Case One: A 37-year-old multipara presented during the 22nd gestational week with ultrasound findings of fetal cardiac enlargement, pericardial effusion, and fetal cardiac hypertrophy in 2019. She was referred because of suspected structural abnormalities of the fetal heart. An ultrasonographic examination during the 23rd gestational week revealed a small-for-gestational-age fetus (estimated fetal weight 532 g, 6.7th percentile) with a cardiac enlargement (Figure 1A) and tachycardia at 172 bpm. The fetal eyeball was prominent with mild proptosis (Figure 1B). The fetal thyroid was diffusely enlarged with abundant blood flow (Figure 1C). She had a history of Graves disease with hypothyroidism after ^131^I treatment in 2016 and was maintained on a levothyroxine supplement. She gave birth to a male infant with low birth weight during the 33rd gestational week in 2018, while the infant was diagnosed with hyperthyroidism with exophthalmos and a high TRAb level at one month of age. Propylthiouracil was administered for 3 months. The family history of the woman was unremarkable. 

Cordocentesis was performed during the 24th gestational week of pregnancy in 2019. The results of karyotyping and chromosomal microarray (CMA) showed negative. Fetal serum TSH was 0.018 μIU/mL, and TRAb was 32.74 IU/L (maternal serum TRAb 149.15 IU/L). Fetal hyperthyroidism was considered. Intrauterine treatment with maternal oral administration of 10 mg of MMI three times per day was initiated after informed consent was obtained. The fetal heart rate returned to the normal range 10 days later. The pericardial effusion resolved, and the cardiothoracic ratio decreased. During treatment, the maternal thyroid function was monitored, and the dosage of levothyroxine was adjusted accordingly (Table 1). Ultrasonography at the 27th gestational week showed a reduction in thyroid hyperemia and slight cerebral ventriculomegaly (10.2 mm).

The mother labored spontaneously and delivered a 1900 g (0.1st centile) female infant at 36^+6^ weeks of gestation. The results of umbilical cord blood showed a high TRAb level of 138.8 IU/L. The newborn had congenital hypoplasia of the laryngeal cartilage and required respiratory support. Six days after birth, the newborn presented with tachycardia, shortness of breath, and irritability. The results of thyroid function tests showed neonatal hyperthyroidism. Methimazole and propranolol were started, with levothyroxine added back due to drug-induced hypothyroidism. Ultrasonography showed abundant blood flow in the goiter, TRAb gradually declined from 94 IU/L (at 6 days), 27 IU/L (at 22 days), 7.5 IU/L (at 35 days), and 1.7 IU/L (at 50 days), to 0.85 IU/L (at 63 days). Drug treatment was sustained for 8 months until the newborn regained euthyroid status with normal growth and development.

Case Two: A 33-year-old multipara who presented with a previous history of Graves disease that was treated with ^131^I in 2010 resulted in normal thyroid function. She had two first-trimester spontaneous miscarriages in 2013 and 2014. In 2016, the woman developed hypothyroidism. While receiving hormone replacement therapy with levothyroxine at 25 μg per day, she was pregnant for the third time in 2016. A morphological scan at 23^+^ weeks of gestation showed fetal tachycardia, an enlarged heart, and oligohydramnios. Three weeks later, intrauterine fetal death occurred, likely due to fetal heart failure. Following induced labor, the female at stillbirth was found to have an enlarged abdomen. The placenta was large. Genetic and pathological investigations were not performed. Their family history was unremarkable.

The woman was transferred to our center because of an ultrasonographic sign of fetal heart failure with suspected congenital heart disease during the 23rd gestational week (in the 4th pregnancy) in 2021. A further ultrasonographic scan showed a severely enlarged heart with a cardiothoracic ratio of 0.66, a pericardial effusion of 4.4 mm, tachycardia with a fetal heart rate of 160–170 bpm, tricuspid regurgitation (145 cm/s), mild fetal ascites, palpebral oedema (Figure 2A–C), oligohydramnios and mild renal hypoplasia. The expected fetal weight was 653 g (63rd centile). Furthermore, the fetal thyroid gland was congested (Figure 2D). A detailed consultation revealed a maternal history of hyperthyroidism followed by hypothyroidism. The thyroid function test showed a euthyroid status with the markedly raised titer of the thyroid-stimulating hormone receptor antibody (TRAb) at 15.27 IU/L (normal range: 0–1.22 IU/L) in the early stage of pregnancy, rising to 103.8 IU/L at 23^+^ weeks of gestation.

Cordocentesis was performed during the 23^+^ gestational week of pregnancy in 2021. The results of karyotyping, CMA, and whole exome sequencing showed negative. Fetal serum Hb was 92 g/L, TSH was 0.09 μIU/mL, FT4 was 8.68 ng/dL, and TRAb was 31.9 IU/L. After multidisciplinary discussion and communication with the parents, we started empirical treatment with 10 mg of methimazole to the mother orally three times per day. The fetal heart rate returned to normal within 3 days. During the 26th gestational week, oligohydramnios, cardiac enlargement, pericardial effusion, and thyroid hyperemia resolved. During fetal anti-thyroid therapy, maternal monitoring showed hypothyroidism (TSH 14.5 μIU/mL, FT4 0.59 ng/dL), and the dose of MMI was reduced with added levothyroxine 100 μg once per day (Table 2). During the 31st gestational week, the cardiothoracic ratio decreased to 0.59 (Figure 2E), and the amniotic fluid volume measurement and Doppler study of the umbilical artery and middle cerebral artery showed normal findings. 

The mother labored spontaneously during the 36th gestational week, delivering a 3000 g female infant with good Apgar scores. A physical examination of the newborn revealed a grade 1 goiter. Ultrasonography showed diffuse enlargement of the thyroid gland with a focal inferno. Blood analysis showed a high TRAb of 103.2 IU/L and TSH of 0.015 μIU/mL with FT4 > 12 ng/dL. Five days after birth, the heart rate fluctuated between 140 and 178 bpm. The diagnosis of congenital hyperthyroidism was confirmed. MMI was administered with propranolol as the neonate had tachycardia (a heart rate up to 193 bpm). Propranolol was stopped on the 30th day after birth. Simultaneously, the neonatal became hypothyroid. The dosage of MMI was gradually reduced, with the addition of levothyroxine at 5 μg per day until they were 1 year old. At the age of two, the baby was examined with normal growth and development (Table 3).

## 3. Discussion

Fetal hyperthyroidism can be classified as autoimmune and non-autoimmune. The former usually occurs secondary to maternal Graves disease. It has been estimated that about 0.2% of pregnant women suffer from Graves disease [3], and in these pregnant women, 1 out of 70 develop fetal thyrotoxicosis [11], a serious condition that can be associated with fetal death or long-term sequelae [12].

The fetal thyroid gland develops as a midline endodermal invagination of the foramen cecum of the tongue. These structures fuse, and the thyroid gland migrates to its definitive position in the anterior neck by the seventh week of embryonic development [13,14]. The bilobed thyroid gland, with thyroid follicles containing colloid, becomes visible by the 10th gestational week. It can extract and concentrate the iodine that reaches the gland through the bloodstream to convert it into thyroxine (T4) and triiodothyronine (T3) [15,16]. Thyroid hormones are essential during pregnancy for fetal neurogenesis [17]. Under normal conditions, the thyroid gland secretes about 80% T4 and 20% T3, as well as thyroglobulin [16]. The fetus relies on the maternal contribution as the major source of thyroid hormones, especially in the first trimester of pregnancy. Around the 10th–12th gestational week, the fetus begins to synthesize thyroid hormones with a low level at the beginning of pregnancy and rising to a significant level starting in the second trimester. The reason for this delay is that the fetal thyrotropin receptor begins to respond to the stimulation of TSH in the second trimester. In spite of the high concentration of TRH, the TSH concentration in the fetal circulation remains low until the thyroid function matures at the 18th–20th gestational week, after which it gradually increases to a peak by the term of gestation. In response to the increasing concentration of TSH, the T4 level increases [18,19,20]. 

Despite the development and maturation of the fetal thyroid, its function is relatively independent of the mother. The transplacental passage of iodine and T4 from the mother to the fetus takes place throughout the gestation period. However, rising T4 levels have minimal inhibition of TSH secretion in utero because of the immaturity of the negative feedback response in the fetal pituitary gland. The hypothalamic–pituitary–thyroid axis does not become fully functional until about 1 to 2 months after birth [21,22]. Therefore, fetuses are prone to thyroid dysfunction due to the impact of maternal thyroid function and their own health status.

TRAb is an IgG antibody that, like iodine and antithyroid drugs, can cross the placenta and reach the fetus. Maternal TRAb levels are the highest in the first trimester and gradually decrease in the second and third trimesters. Ten percent of the maternal level is transferred to the fetus in the 17th–22nd gestational week, while 50% is transferred to the fetus in the 28th–32nd gestational week [23,24]. It does not reach the maternal level until the third trimester with an increase in placental permeability, and the TSH receptor does not react to TRAb until the second trimester. TRAb can be transferred from the mother to the fetus and stimulates the adenylate cyclase in fetal thyrocytes, leading to excessive thyroxine secretion and causing thyrotoxicosis in utero [15]. Therefore, fetal hyperthyroidism occurs more often in the second half of pregnancy [25].

In early pregnancy, enquiring about the maternal history of thyroid disease is important to facilitate proper management and the early detection of fetal hyperthyroidism. In this report, both cases had a history of Graves disease and received ^131^I treatment. However, the clinical manifestation was hypothyroidism rather than hyperthyroidism during pregnancy, for which levothyroxine replacement therapy was used, and the maternal TRAb level simultaneously increased. This highlights the significance of inquiry on previous history, regardless of hypothyroidism or hyperthyroidism. A randomized controlled trial showed that in women treated with radioiodine ablation (RAI; ^131^I) or total thyroidectomy, increased TRAb levels could persist for several years after definitive therapy [26]. Some patients with GD had a transient TRAb increase 6 months after ^131^I treatment, which might be mediated by the release of thyroid antigens from damaged thyrocytes [27]. Takamura et al. [28] showed that it took about 3 years to normalize the TRAb value after total thyroidectomy. Therefore, the level of TRAb should be accessed in patients with a history of GD. The Guidelines for Diagnosis and Treatment of Hyperthyroidism and Other Causes of Thyrotoxicosis, published by the American Thyroid Association in 2016, recommends that GD patients should be tested for TRAb at the early stages of pregnancy. If the TRAb level increases, the TRAb should be retested at the 18th–22nd gestational week. Women with elevated TRAb levels at the 18th–22nd gestational week should have TRAb remeasured in late pregnancy (the 30th–34th gestational week) to guide decisions regarding neonatal monitoring [29]. 

Although it has been proposed that the clinical picture may start from the 21st gestational week when the antibody level increases and the fetal TSH receptors become responsive, the picture of fetal hyperthyroidism generally becomes prominent in the 26th–28th gestational week [16]. In this report, the two cases progressed to a relatively severe degree earlier in the 22nd–23rd gestational week, with significant fetal hyperthyroid symptoms of heart failure. This earlier onset of symptoms may be related to higher TRAb levels, up to 30 IU/L in the fetal blood. van Dijk [30] proposed that the lowest level of maternal TRAb in neonatal hyperthyroidism measured with a second-generation assay is 4.4 U/L, which corresponds to 3.7 times the upper limit normal. The long duration of neonatal treatment for the two babies may also be related to the TRAb level. Zakarija M [31] mentioned that after birth, maternal antibodies gradually decrease and disappear from the infant’s circulation by 4 months of age at most and, on average, by 1 month of age. However, the newborns of our two cases were shown to have increased TRAb levels for more than 4 months. 

Goiter is the earliest sonographic sign of fetal thyroid dysfunction and an unambiguous indication of fetal thyroid dysfunction [31,32,33]. Fetal goiter has an incidence of 1:40,000 deliveries [34]. Many studies describe how, in the Doppler ultrasound, if the blood supply is more intense in the center, the possibility of hyperthyroidism is higher. If the blood supply is more intense in the periphery, the possibility of hypothyroidism is higher [35]. Luton et al. [32] showed that fetal goiter indicates abnormal thyroid function, with a sensitivity of 92%, a specificity of 100%, a positive predictive value of 100%, and a negative predictive value of 98%. Ranzini et al. [36] reported data on the normal size range of the fetal thyroid gland by gestational age. In our experience, detecting a goiter is not easy in the prenatal period, although it is the earliest alarming signal. Fetal tachycardia is another alarming signal for hyperthyroidism, but it occurs later than the fetal goiter. In this report, both fetuses presented with tachycardia in early gestation, accompanied by signs of heart failure such as cardiac enlargement and pericardial effusion. These signs prompted us to look for other signs of fetal hyperthyroidism. The integration of fetal ultrasound and Doppler for the surveillance of the size and blood flow signals of the fetal thyroid may enhance early detection and reduce the missed diagnosis of fetal hyperthyroidism. 

Fetal bone age was interpreted as the presence of knee epiphytic nuclei. Distal femoral epiphyses become prominent at about the 32nd gestational week. The appearance of distal femoral epiphyses before the 31st gestational week is considered an advanced bone age, and the absence of the distal femoral epiphyses at the 33rd gestational week is considered a delayed bone age [37]. Fetal hyperthyroidism usually manifests as advanced bone maturation. Between 31 and 33 weeks of gestation, ultrasonography can detect advanced bone maturation in fetuses with hyperthyroidism. 

The first child developed laryngomalacia at birth. Multiple causal theories of laryngomalacia have been proposed. Neurologic dysfunction is one of the leading theories, suggesting an altered laryngeal tone due to the abnormal integration of the laryngeal nerves. Thyroxine plays a crucial role in the neural development of fetuses, but its relationship with hyperthyroidism is unknown [38].

Fetal hyperthyroidism might also be associated with intrauterine growth restriction, both from the direct effect of hyperthyroidism and associated pre-eclampsia [39]. A preterm birth is more common if fetal hyperthyroidism is left untreated [40]. Preterm delivery occurs in 4–11% of mothers treated for thyrotoxicosis during pregnancy and in 53% of mothers who remain untreated [41]. In this report, both women had a preterm birth, and Case One had early-onset fetal growth restriction. 

The definitive prenatal diagnosis of fetal hyperthyroidism depends on testing free T4, free triiodothyronine (T3), and TSH levels in the fetal blood. However, fetal cordocentesis is only performed under limited circumstances, as it has a certain probability of leading to serious complications [42]. Since most ultrasonographic manifestations are not specific to fetal hyperthyroidism, a review of family history and genetic analysis should be conducted to exclude possible genetic causes of fetal cardiac enlargement. 

Non-autoimmune hyperthyroidism (OMIM 609152) is rare. It can be caused by an activated mutation in the thyroid-stimulating hormone receptor (TSHR) gene [43,44]. The TSHR gene belongs to the G-protein coupled receptor superfamily, which stimulates the growth of thyroid follicular cells and thyroxine synthesis by activating adenylate cyclase. The activation of mutations can lead to neonatal hyperthyroidism, which may be autosomal-dominant or sporadic [26]. Lourenço et al. reported a neonatal case diagnosed with McCune–Albright syndrome (OMIM 174800) and carried a mosaic somatic activating mutation in the alpha subunit of the guanine nucleotide-binding protein (G protein) (GNAS) [44,45].

Fetal hyperthyroidism can be safely and effectively treated by administering antithyroid drugs to the mother. Sometimes, the dosage of antithyroid drugs required to control hyperthyroidism in the fetus leads to hypothyroidism in the mother, in which case thyroxine should be given to the mother [29]. For the treatment of hyperthyroidism in pregnant women, propylthiouracil is preferred to methimazole or carbimazole because the latter has been associated with aplasia cutis congenita and other malformations [46], such as esophageal or posterior nasal atresia, skin aplastic lesions, and embryopathy (including developmental delay, hearing loss, and facial deformities).

However, the best drug for the intrauterine treatment of fetuses has not yet been determined. As mentioned previously, the clinical picture of fetal hyperthyroidism may start from the 21st gestational week. While the teratogenic effect of methimazole is relatively significant in the first three months of pregnancy, the hepatotoxic effect of propylthiouracil in pregnant women continues to exist, and PTU is also associated with leukopenia [47,48]. In the reported cases, we used methimazole as intrauterine treatment, and both fetuses had promising positive effects, with the slowing of fetal tachycardia relatively quickly, subsequently resolving fetal heart enlargement, pericardial effusion, and thyroid hyperemia. Assessing the impact of intrauterine therapy on fetal hyperthyroidism remains challenging. The close monitoring of fetal heart rate, fetal growth, liquor, and Doppler studies are fundamental for assessing the response to the treatment and optimizing the dosage. 

## 4. Conclusions

Fetal hyperthyroidism must be considered if there are ultrasonographic signs of tachycardia, cardiomegaly, pericardial effusion, or intrauterine growth restriction, especially when there is an increase in the maternal TRAb level and fetal goiter. Maternal Graves disease presenting with hypothyroidism might mask the condition. Intrauterine therapy with maternal oral methimazole with the monitoring of the fetal heart rate, fetal cardiac size, and fetal thyroid hyperemia can be effective in treating fetal hyperthyroidism.

## Figures and Tables

**Figure 1 diagnostics-14-00102-f001:**
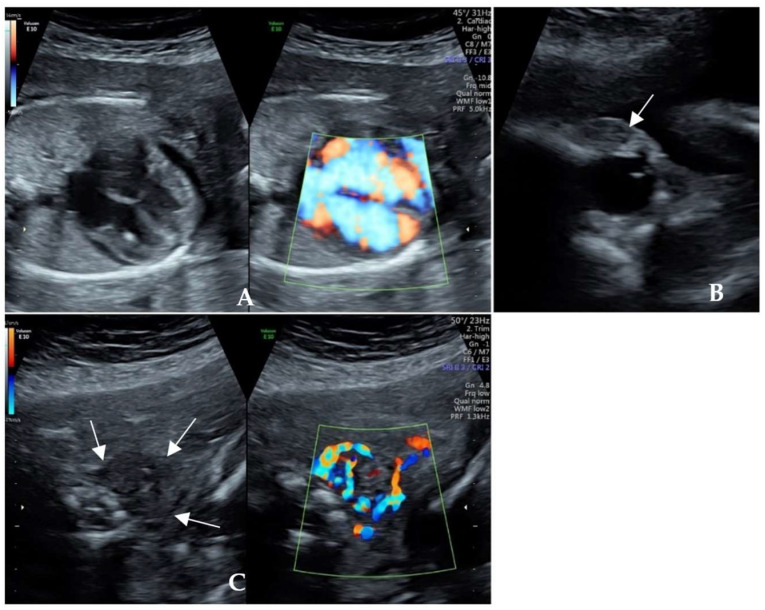
Ultrasonographic cardiac scan of Case One during the 23rd gestational week. (**A**) Fetal cardiac enlargement, pericardial effusion, and cardiac hypertrophy; (**B**) Fetal mild exophthalmos and palpebral oedema; (**C**) Enlarged thyroid with diffusely abundant blood flow.

**Figure 2 diagnostics-14-00102-f002:**
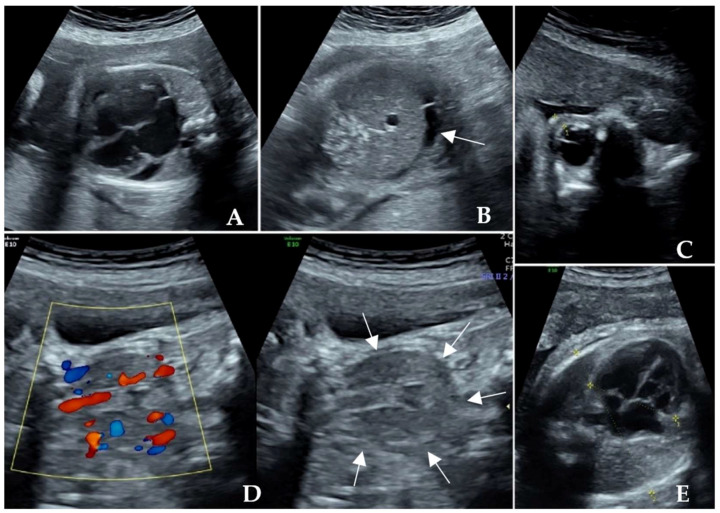
(**A**) Severely enlarged heart with increased cardiothoracic ratio and pericardial effusion at the 23rd gestational week in Case Two; (**B**) Mild fetal ascites; (**C**) Mild fetal palpebral oedema; (**D**) Fetal thyromegaly and congestion; (**E**) Decreased fetal cardiothoracic ratio at 31 weeks.

**Table 1 diagnostics-14-00102-t001:** Thyroid function tests of the mother during pregnancy in 2019 (Case One).

Gestational Weeks	TT4 (μg/dL)	FT4 (ng/dL)	TSH (μIU/mL)	TRAb (IU/L)	Treatment	
					LT4 (μg)	MMI (mg)
24^+0^	/	/	0.51	103	/	10 (BID)
24^+6^	7.7	0.85	3.252	/	/	10 (BID)
25^+6^	5.2	0.59	14.5	/	100 (QD)	10 (BID)
27^+4^	8.6	0.82	11.473	/	100 (QD)	10 (BID)
28^+6^	16.5	1.16	1.185	/	100 (QD)	10 (BID)
32^+1^	11.4	0.97	0.713	/	75 (QD)	10 (QD)
34^+0^	12.7	1.03	0.772	/	75 (QD)	10 (QD)
36^+0^	11.5	0.98	1.553	/	75 (QD)	10 (QD)

TT4: total thyroxine; FT4: free thyroxine; TSH: thyroid-stimulating hormone; TRAb: thyroid-stimulating hormone receptor antibody; LT4: levothyroxine sodium; MMI: methimazole; BID: Twice a day; QD: once a day.

**Table 2 diagnostics-14-00102-t002:** Thyroid function tests of the mother during pregnancy in 2021 (Case Two).

Gestational Weeks	TT4 (ug/dL)	FT4 (ng/dL)	TSH (μIU/mL)	TRAb (IU/L)	Treatment	
					LT4 (μg)	MMI (mg)
25^+2^	17.2	/	0.014	149.15	100 (QD)	10 (BID)
26^+1^	17	1.27	0.02	/	100 (QD)	10 (BID)
27^+1^	16.6	1.13	0.029	/	100 (QD)	25 (QD)
28^+1^	14.6	1.19	0.071	/	100 (QD)	25 (QD)
31^+2^	14.8	1.12	0.558	167.1	125 (QD)	10/15 (Alt D)
34^+0^	15.4	1.34	0.167	/	100/125 (Alt D)	20/15 (Alt D)
36^+2^	17	1.21	0.353	/	100 (QD)	20/15 (Alt D)

TT4: total thyroxine; FT4: free thyroxine; TSH: thyroid-stimulating hormone; TRAb: thyroid-stimulating hormone receptor antibody; LT4: levothyroxine sodium; MMI: methimazole; BID: twice a day; QD: once a day; Alt D: alternate days.

**Table 3 diagnostics-14-00102-t003:** Comparison of the two cases of fetal hyperthyroidism with maternal hypothyroidism and intrauterine therapy.

	Case One	Case Two
Fetal manifestations	Fetal cardiac enlargement, pericardial effusion and cardiac hypertrophy, FGR, tachycardia, proptosis, and ventriculomegaly	Fetal tachycardia, enlarged heart and pericardial effusion, tricuspid regurgitation, oligohydramnios, and minor renal dysplasia
Weeks of gestation at presentation	22^+^ weeks	23^+^ weeks
Maternal thyroid status	Hypothyroidism (maintained with 75 μg of levothyroxine supplement), TRAb of 149.15 IU/L	Hypothyroidism (maintained with 25 μg of levothyroxine supplement), TRAb of 103.8 IU/L.
Previous pregnancy	G2P1 (Preterm delivery of a male infant during the 33rd gestational week in 2018, which was small-for-gestational-age with no other clinical manifestation; one month later, they were diagnosed with hyperthyroidism.)	G4P0 (two first-trimester spontaneous miscarriages. The third pregnancy showed fetal tachycardia, an enlarged heart, and oligohydramnios at 23^+^ gestational weeks; intrauterine fetal death occurred three weeks after.)
Previous history	Graves disease, treated with ^131^I	Graves disease, treated with ^131^I
Fetal thyroid status	Thyroid diffusely enlarged with abundant blood flow, hyperthyroidism (TSH 0.018 μIU/mL and TRAb 32.74 IU/L)	Fetal thyroid gland congested, hyperthyroidism (TSH 0.09 μIU/mL and TRAb 31.9 IU/L)
Fetal genetic test	CMA -ve, karyotype -ve	Trio-WES -ve, CMA -ve, karyotype -ve
Intrauterine treatment	Maternal oral administration of methimazole from 22^+^ weeks of gestation	Maternal oral administration of methimazole from the 23^+^ week of gestation.
Significant signs of response	The fetal heart rate returned to the normal range 10 days later. The pericardial effusion and cardiothoracic ratio were relieved.	The fetal heart rate returned to normal at the 26th gestational week. Oligohydramnios, cardiac enlargement, pericardial effusion, and thyroid hyperemia were resolved.
Conditions of delivery	Spontaneous delivery at 36^+6^ weeks of gestation, 1900 g	Spontaneous delivery at 36 weeks of gestation, 3000 g
Outcome of the newborn	Neonatal hyperthyroidism, alive and healthy	Neonatal hyperthyroidism, alive and healthy

Abbreviations: FGR, fetal growth restriction; -ve, negative; CMA, chromosomal microarray analysis; WES, whole exome sequencing.

## Data Availability

The authors declare that the data for this research are available from the correspondence authors upon reasonable request.

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
