# Peer review of "Fetal Hyperthyroidism with Maternal Hypothyroidism: Two Cases of Intrauterine Therapy"

_diagnostics, 2024, doi:10.3390/diagnostics14010102_

Round 1

Reviewer 1 Report

Comments and Suggestions for Authors

Dear Authors,

The manuscript entitled “Fetal Hyperthyroidism with Maternal Hypothyroidism: Two Cases of Intrauterine Therapy and Literature Review” shows two cases of Graves’ disease in pregnant women, with a particular affectation of the fetus and the need of intrauterine treatment.

They are 2 clinical case reports of huge interest, that have been described in detail, with complementary images of sonographic findings and extended follow-up of mother-child pairs. The discussion is very well written and summarizes the most pertinent information on this field.

I have just a few comments for the authors:

1.      I miss some references such as:  Management of thyrotoxicosis during pregnancy.

Andersen SL, Knøsgaard L. Best Pract Res Clin Endocrinol Metab. 2020 Jul;34(4):101414. doi: 10.1016/j.beem.2020.101414.

2.      It is shocking to find 5 references from the same author: Polak M.

3.      Line 91: If the two miscarriages did happen in 2014, it would sound better to say: In 20214, she had 90 two first-trimester spontaneous miscarriages, a few months apart.

Author Response

Dear Editors and Reviewers:

Thank you for your comments concerning our manuscript entitled " Fetal Hyperthyroidism with Maternal Hypothyroidism: Two Cases of Intrauterine Therapy and Literature Review” (diagnostics-2733354). The comments are very valuable and  helpful for revising and improving the manuscript. We hope the corrections would meet with your approval.

  • Comments 1: I miss some references such as: Management of thyrotoxicosis during pregnancy. Andersen SL, Knøsgaard L. Best Pract Res Clin Endocrinol Metab. 2020 Jul;34(4):101414. doi: 10.1016/j.beem.2020.101414.

Response 1:Thank you for alerting us to the publication. We have quoted the reference in the introduction (line 279-281) and included this in the reference list.

  • Comments 2: It is shocking to find 5 references from the same author: Polak M.

Response 2: Thank you for pointing this out. Polak M is a very outstanding scholar in this area.

  • Comments 3: Line 91: If the two miscarriages did happen in 2014, it would sound better to say: In 20214, she had 90 two first-trimester spontaneous miscarriages, a few months apart.

Response 3: Thank you for pointing this out. We have updated the history and stated that the patient had two first-trimester spontaneous miscarriages in 2013 and 2014 respectively (line 95).

Reviewer 2 Report

Comments and Suggestions for Authors

You describe two cases of fetal hyperthyroidism with maternal hypothyroidism and reviewed the assessment and intrauterine therapy of fetal hyperthyroidism. The development of such fetal hyperthyroidism in women treated with radioactive iodine previously for Graves hyperthyroidism is well-known and should be avoided nowadays. The cases have nothing special in the development of fetal hyperthyroidism, may be the diagnosis earlier than it is described in the literature. The subject is interesting, especially, because these cases are quite rare and it is very difficult to recognize and to treat fetal hyperthyroidism. In my opinion, you must emphasize more on the therapy and results of this therapy.

Author Response

Dear Editors and Reviewers:

Thank you for your comments concerning our manuscript entitled " Fetal Hyperthyroidism with Maternal Hypothyroidism: Two Cases of Intrauterine Therapy and Literature Review” (diagnostics-2733354). Those comments are very valuable and helpful for revising and improving our paper. We have updated the manuscript and hope that the corrections would meet with your approval.

  • Comments 1: Title – The title is suggestive for the topic of the article, but no literature review was done when the cases are analyzed. No other similar cases were used to compare their own cases. I suggest to replace literature review from the title or to find some other similar cases. Maybe you should refer to the American guideline: 2017 Guidelines of the American Thyroid Association for the Diagnosis and Management of Thyroid Disease During Pregnancy and the Postpartum.

Response 1: Thank you for pointing this out. We have updated the title to “Fetal Hyperthyroidism with Maternal Hypothyroidism: Two Cases of Intrauterine Therapy” and reviewed the literature in the discussion section. We have also included the reference on American guideline: 2017 Guidelines of the American Thyroid Association for the Diagnosis and Management of Thyroid Disease During Pregnancy and the Postpartum.

  • Comments 2: The abstract is informative enough.

Response 2: Thank you for your comment.

  • Comments 3: Keywords –I suggest to avoid abbreviation in keywords – TRAB.

Response 3: Thank you for pointing this out. We have changed “TRAb” to “ thyroid-stimulating hormone receptor antibody (TRAb)” in keywords.

  • Comments 4: The introduction presents the known data on this subject and the reason to present these cases. I suggest to add some information about the recommended management of Graves’ disease in young women to avoid the development of fetal hyperthyroidism.

Response 4:  Thank you for the suggestion. We have referred to the 2017 American guideline and added information about the recommended management of Graves disease in women for early detection of fetal hyperthyroidism in the introduction:

“For women with current or past Grave’s disease, assays to measure TRAb should be performed at the beginning of pregnancy, with fetal ultrasound surveillance for early detection of fetal hyperthyroidism”(line50-52)

  • Comments 5: Case presentation: the descriptions are like a story, without concrete data of lab determinations, normal values etc. I suggest to add hormonal values during follow-up of pregnant women, in addition to the therapy with thyroxine and antithyroid drugs. May be a table with this value could be useful. In some parts the description is not very clear: For ex. - row 120: ”The mother labored spontaneously at 36th week of gestation, delivering a 3000g female infant with good Apgar scores. Physical examination revealed grade 1 goiter.” Physical exam of newborn or of mother? It should be specified.

Response 5: Thank you for pointing these out.

  • We have added two tables to show the fluctuation of maternal thyroid function with adjustment of maternal medical treatment (table 1 and table 2).
  • “Physical exam of newborn or of mother” : Physical examination was on the newborn. We have modified the sentence to “Physical examination of the newborn revealed grade 1 goiter”(line 126).
  • Comments 6: The discussions address the subject of the paper, but they are too long and a comparison with other similar cases should be more interesting. Indeed, the strong aspects of this article are the early prenatal diagnosis and the successful treatment to avoid fetal death, as a consequences of untreated fetal hyperthyroidism.

Response 6: Thank you for your comment. It is a very well suggestion, but we just try to focus on the reviews of fetal hyperthyroidism at the beginning of the manuscript, instead of the comparison with other case.

Comments 7: Conclusions should be reformulated: ”Fetal hyperthyroidism should be excluded if there are ultrasonographic signs of tachycardia, cardiomegaly, pericardial effusion or intrauterine growth restriction, especially when there is raised maternal TRAb level and fetal goiter” I suggest to replace ”should be excluded” with ”must be taken into account” It is about maternal treated Graves’ disease, and the therapy is represented by methimazole added to levothyroxine therapy. The conclusions should be adjusted to be much more accurate.

Response 7: Agree. We have revised “should be excluded” to “must be taken into account” to emphasize this point.  

  • Comments 8: The references are written according to the instructions for authors and are relevant for the subject of this article.

Response 8: Thank you for your comment.

Response to Comments on the Quality of English Language

Point 1: The English language is good, in my opinion, but I am not a native English speaker. May be in the discussions part of this article, the understanding of the text is quite difficult. Some mistakes are present, for ex in rows 102-103:  (Figure 2A, B, C), ol-igohydramnios

Response 1: Thank you for your comments. We shall ask if editors could format the pages so that words are not hyphenated when sentence is carried to next line, and “ol-igohydramnios” can appear as ‘oligohydramnios’.

Round 2

Reviewer 2 Report

Comments and Suggestions for Authors

Thank you for your response. I agree with you, that the editors has to format the page according to your request. The tables you inserted are according to our request, but they must be placed in the middle of the page. 

In the row 182 - ”TRAb cannot completely disappear from the maternal circulation following the treatment with I131 or thyroidectomy” - The TRAB usually disappear from maternal circulation after total thyroidectomy, this is why TT is the desired therapy for women before a pregnacy - see bibliography:

Takamura Y, Nakano K, Uruno T, Ito Y, Miya A, Kobayashi K, Yokozawa T, Matsuzuka F, Kuma K, Miyauchi A. Changes in serum TSH receptor antibody (TRAb) values in patients with Graves' disease after total or subtotal thyroidectomy. Endocr J. 2003 Oct;50(5):595-601

Yoshioka W, Miyauchi A, Ito M, Kudo T, Tamai H, Nishihara E, Kihara M, Miya A, Amino N. Kinetic analyses of changes in serum TSH receptor antibody values after total thyroidectomy in patients with Graves' disease. Endocr J. 2016;63(2):179-85. 

Author Response

Many thanks for your comments, we have deleted“TRAb cannot completely disappear...”, and added "Some patients with GD had a transient TRAb increase 6 months after 131I treatment, ..." in the row 253-256. And we have inserted the table in the middle of the page.